# Is There Any Improvement of the Coagulation Imbalance in Sickle Cell Disease after Hematopoietic Stem Cell Transplantation?

**DOI:** 10.3390/jcm8111796

**Published:** 2019-10-26

**Authors:** Laurence Rozen, Denis F. Noubouossie, Laurence Dedeken, Phu Quoc Lê, Alina Ferster, Anne Demulder

**Affiliations:** 1Laboratory of Hematology LHUB-ULB ULB Université Libre de Bruxelles, 1020 Brussels, Belgium; noubouossie75@yahoo.fr (D.F.N.); Anne.DEMULDER@LHUB-ULB.BE (A.D.); 2Hematology Oncology Unit, Hôpital and niversitaire des Enfants Reine Fabiola, ULB Université Libre de Bruxelles, 1020 Brussels, Belgium; laurence.dedeken@huderf.be (L.D.); phuquoc.le@huderf.be (P.Q.L.); alina.ferster@huderf.be (A.F.)

**Keywords:** sickle cell disease, hematopoietic stem cell transplantation, thrombin generation test, coagulation, hemostatic potential

## Abstract

Several components of the clotting system are modified towards hypercoagulability in sickle cell disease (SCD). To date, hematopoietic stem cell transplantation (HSCT) is the only validated curative treatment of SCD. Here, we investigated the changes in the hemostatic potential of SCD children who’ve received a successful HSCT. Seventeen children with severe SCD were enrolled in the study. Thrombin generation (TG) was performed on citrated platelet-poor plasma, obtained before and 3, 6, 9, 12 and 15 months after HSCT. TG was triggered using 1 pM tissue factor and 4 µM phospholipids with or without thrombomodulin (TM). Before the HSCT, SCD children showed a higher endogenous thrombin potential (ETP), higher peak, higher velocity and shorter time-to-peak of TG than the normal controls (NC). ETP did not significantly change following the HSCT. However, the peak, velocity and time-to-peak of TG reversed to normal ranges from 3 months post-HSCT and remained so up to 15 months post-HSCT. The reduction of ETP after the addition of thrombomodulin (RETP) was dramatically reduced in SCD children before HSCT as compared with the NC. A partial reversal of RETP was observed from 3 months through 15 months post-HSCT. No statistical difference was observed for patient age or donor hemoglobinopathy status. In summary, successful HSCT improves the kinetics of TG but not the total thrombin capacity in SCD children.

## 1. Introduction

Sickle cell disease (SCD) is considered as a hypercoagulable state. Clinical manifestations include venous thromboembolism and stroke in a subset of patients [1,2]. The activation of coagulation is often present, with consumptive coagulopathy as well as increased levels of in vivo markers of thrombin and fibrin generation [3,4].

Several components of the clotting system are modified towards hypercoagulability in SCD. Previous studies have reported that patients with SCD present with alterations of several of the individual procoagulant and anticoagulant factors involved in the coagulation system (may it be pro or anticoagulants: elevated FVIII levels, decreased protein C, and protein S), but also an increase in the hemostatic potential using global tests of coagulation. Global tests reflect a better hemostatic balance than the usual test performed in a routine laboratory; they are the final result of the interaction between pro- or anti-coagulant factors. As thrombin is a central factor of the coagulation cascade, the evaluation of the thrombin generation (TG) is a key element in the evaluation of coagulation disorders and easy to measure thanks to the Calibrated Automated Method (CAT^®^), developed by Hemker [5]. This method provides a curve characterized by an initiation phase (lag-time, corresponding to the clotting time of the routine tests) followed by an amplification phase reaching a peak of maximal thrombin formation. This phase is followed by a decay phase, corresponding to the intervention of anticoagulant factors. ETP (the area under the curve), represents the total amount of thrombin formed as a result of pro- and anticoagulant factors. Velocity represents the rate of thrombin formation, and time-to-peak the time to reach the peak of thrombin formation. Hypercoagulable states, as is the case in SCD, show a reduced lag-time, and increased ETP and peak [6]. Therefore, observations with global tests have demonstrated in children with SCD that, even at a steady state, hypercoagulability is present [7,8]. Elevated thrombin generation (TG) in SCD children has been described as being related to younger age as well as to the intensity of hemolysis and a link between TG and cerebral vasculopathy in these patients has been mentioned [9].

The treatment of SCD patients has been considerably improved with the introduction of hydroxyurea (HU) and with much better global medical management, at least in developed countries [10,11]. However, SCD remains associated with severe morbidity and decreased survival (organ failure, chronic damage). To date, matched sibling donor (MSD) hematopoietic stem cell transplantation (HSCT) is the only validated curative treatment of SCD [12].

HSCT is mainly considered for children with severe clinical disease characterized by a history of stroke, repeated vaso-occlusive crises (VOC), multiple acute chest syndrome (ACS), and multiple osteonecrosis or allo-immunization that hampers the appropriate management of acute or chronic complications [13]. Patients with deteriorating lung or kidney disease also benefit from transplantation [14] as well as children at risk of stroke [15].

The overall survival (OS) and event-free survival (EFS) after MSD-HSCT ranges from 93% to 97% and from 82% to 86% [16,17,18,19,20]. Despite these impressive clinical results, the evolution of the hemostatic balance following HSCT is still unknown.

According to a recent study [21], HSCT affects the coagulation system mainly in the first few weeks after transplantation. This study reports a clear shift towards a procoagulant state with an increase of the coagulation activation markers (Von Willebrand factor, pro-thrombin fragments 1 + 2), fibrin turnover (D-dimers), and ETP. These early coagulation abnormalities can lead to an increased morbi-mortality rate [22], with transplant-associated thrombotic microangiopathy (TA-TMA) as a major side effect, mainly due to endothelial damage [23,24].

As SCD and HSCT confers a prothrombotic profile to patients, the aim of our study was to investigate whether the hemostatic imbalance in SCD children could improve following HSCT. We used the Calibrated Automated Thrombogram^®^ to compare the hemostatic potential of SCD children before and after HSCT.

## 2. Materials and Methods

### 2.1. Patients

From November 2014 to March 2018, children with severe SCD undergoing MSD-HSCT were enrolled in the study. Before HSCT, all patients were suffering from severe clinical diseases, including cerebrovascular disease, recurrent ACS or frequent VOC (≥2/year).

### 2.2. Controls

The population of normal controls (NC) has already been described in a previous publication [25] and consisted of 23 children aged from 2 to 14 years who had a blood coagulation test before a minor elective surgery. The remaining plasmas of those samples were used as controls for the present study.

### 2.3. Blood Collection and Handling

The study was approved by the local Ethics Committee of the Hôpital Universitaire des Enfants Reine Fabiola (n° 25/09) and informed consent was obtained from each parent or legal tutor before transplant. The peripheral venous blood of patients and normal controls (NC) was collected into Vacutainer^®^ tubes (BD, Plymouth, UK) containing buffered sodium citrate (0.109M). Venipuncture was performed using a butterfly 21G needle. Platelets-poor plasma was prepared within the hour after blood collection by a double centrifugation at 2500× *g* for 15 min. Samples were stored at −80 °C with an average time of between 6 and 12 months. Prior to analysis, samples were rapidly thawed for 5 min in a water bath at 37 °C.

### 2.4. Thrombin Generation Assay

TG was performed on citrated platelet-poor plasma, obtained before and 3, 6, 9, 12 and 15 months after HSCT using the CAT^®^ method after the addition of 1 pM tissue factor and 4 µM phospholipids without and with thrombomodulin TM (PPP low +/− TM, Synapse BV). The detailed protocol has already been described by Noubouossie et al. [7,9].

Internal quality control was ensured by performing reference plasma (Poolnorm, Diagnostica Stago, Asnières, France) on each TG plate.

### 2.5. Evaluation of Hemolysis and HbS

These tests were performed in the samples collected for routine follow-up, in parallel with those collected for thrombin generation before HSCT, 6 and 15 months after HSCT. The rate of hemolysis was evaluated in SCD patients using total hemoglobin level (HB and plasma lactate deshydrogenase level LDH. The total hemoglobin level was measured in the blood collected in dipotassic EDTA tubes (Vacutainer^®^, Becton Dickinson, Plymouth, UK) using an automate cell counter (Sysmex XN 9000 Sysmex Corporation TM, Kobe, Japan). The LDH levels were measured in the plasma collected in lithium heparin tubes with a gel separator (Vacutainer^®^, Becton Dickinson, Plymouth, UK) using the LDHI2 on Cobas 8000^®^ analyzer (Roche Diagnostics, Rotkreuz, Switzerland).

HbS was assessed by alkaline capillary electrophoresis (Capillarys systems, Sebia, Benelux).

### 2.6. Statistical Analysis

The TG parameters (endogenous thrombin potential (ETP), peak, velocity, time to peak, lag time) were measured. The reduction of ETP (RETP) in the presence of TM was calculated for each sample and expressed in percentage. Patient results were expressed as median with a range at each time point and compared with the range (percentile 2.5–97.5) of normal controls (NC).

The Mann–Whitney test was used to compare the NC and patients before HSCT and to compare TG for patients <10 years and >10 years and to compare TG after 15 months regarding AA versus AS donor.

The TG parameters were compared before and 3 months after HSCT using the Wilcoxon matched pairs test.

Hb, HbS and LDH values were compared before HSCT and 6 and 15 months after HSCT with the Friedman test. In the cases where Friedman tests were significant, Dunn’s post test was performed to compare all pairs of the groups.

The statistical calculations were realized using the software GraphPad Prism version 5 (Graph-Pad Software Inc., San Diego, USA). A *P* value less than 0.05 was considered significant.

## 3. Results

### 3.1. Demographic Data

Seventeen children (five females, 12 males) were consecutively enrolled. Sixteen patients were homozygote HbSS and one was compound heterozygote HbSB+. Three patients were on a chronic exchange transfusion program before HSCT, either for cerebro-vascular disease (*N* = 1) or recurrent ACS despite treatment with HU (*N* = 2). All the patients were treated with HU. The median age at transplantation was 9.6 years (range: 3.6–16.5 years). The stem cell donor was a matched sibling in all cases (11 were heterozygous AS, five were AA and one was heterozygous AC). The source of the stem cell was cord blood in one patient, bone marrow in 11 patients and bone marrow plus cord blood in five patients. Table 1 summarizes the demographic data.

All the patients received myeloablative conditioning regimen (Busulfan, Cyclophosphamide and Anti-thymocytes Globulins). The graft-versus-host disease (GVHD) prophylaxis included cyclosporine and methotrexate or cyclosporine and mycophenolate in the case of cord blood.

### 3.2. Clinical Data

All patients engrafted successfully. No acute GVHD occurred. One patient developed chronic extensive GVHD which was well controlled after steroids and extracorporeal photopheresis and could be stopped after five months. After a median follow-up of three years, all patients were alive with a full donor chimerism. Table 1 summarizes the clinical data.

### 3.3. Thrombin Generation Parameters

In the presence or absence of thrombomodulin, the ETP, peak and velocity of TG were significantly higher; lag time and time-to-peak were shorter, whereas the RETP was decreased in children with SCD before HSCT as compared with NC. As it represented a more complete situation, results with TM are illustrated in Figure 1. All values for the TG parameters are summarized in Appendix A.

At 3 months post-HSCT, peak and velocity significantly decreased while lag time and time-to-peak significantly increased as compared with pre-HSCT values, regardless of the presence or absence of TM. The median peak, velocity and time-to-peak returned to values within the normal range, set as (P2.5–P97.5) of the NC (see Figure 2 for results with TM). The lag time and RETP significantly increased at 3 months post-HSCT as compared with pre-HSCT. However, their median values did not return within the normal range, indicating a partial recovery (Appendix A).

From three months through 15 months after HSCT, no further changes were observed in all TG parameters. Median ETP value remained higher while median RETP and median lag time values remained lower than P2.5–P97.5 of NC. In contrast, median values of the peak, time-to-peak and velocity of TG performed in the presence of TM stabilized within the P2.5–P97.5 of NC, suggesting a full recovery (Table 1 and Figure 2, for results with TM).

Otherwise, no statistical difference was observed for patient age or donor hemoglobinopathy status (data not shown but all *p*-value were >0.05).

### 3.4. Evolution Ofhemoglobin, HbS and LDH

Median hemoglobin, HbS and LDH are summarized in Table 1. They change significantly after HSCT. When compared to pre HSCT values, at 6 months, total hemoglobin increased from 8.3 g/dL (range: 7.2–11.4) to 11.7 g/dL (range: 10.5–14.5) (*p* < 0.01). HbS and LDH decreased from 70% (range: 26–88) to 32% (range: 0–40) (*p* < 0.001) and from 466 UI/L (range: 256–823) to 243 UI/L (range: 185–488) (*p* < 0.01), respectively. The comparison of 15 months before and after HSCT showed similar results while the comparison between 6 and 15 months after HSCT did not show any significant changes.

These observations were similar even when not taking into account the three patients under exchange transfusion before HSCT.

## 4. Discussion

To our knowledge, this is the first study addressing the changes in the clotting system following HSCT. As expected, SCD patients in this study presented a hypercoagulable state as compared to the NC before HSCT. This hypercoagulable state included a faster time to generate thrombin (lag time and time-to-peak are shorter in SCD than in NC), a faster rate of thrombin production (velocity is higher in SCD than in NC), a higher concentration and total capacity to generate thrombin (peak and ETP are higher in SCD than in NC). The altered global coagulation in patients with SCD before HSCT was previously highlighted by our team in a limited cohort [26] and by Gerotziafas et al. [8]. Several synergistic factors have been suggested to explain the increased hemostatic potential in patients with SCD. Indeed, we and others have previously reported the decreased levels of protein C and protein S in these patients [26,27,28,29,30,31]. Protein C and protein S are known determinants of TG [32] and decreased levels of these proteins induce a resistance to thrombomodulin [33]. In addition, elevated FVIII levels in these patients inversely correlate with RETP and may contribute to some resistance to activated protein C [34].

Circulating levels of pro-coagulant microvesicles deriving from blood and endothelial cells have also been reported in patients with SCD [35,36,37,38]. Some of these microvesicles, especially those released by activated monocytes or endothelial cells, expose functional tissue factor that may contribute to shortening the lag time and the time-to-peak of TG in these patients [36]. Procoagulant microvesicles also expose phosphatidylserine, a procoagulant phospholipid that provides the surface for the assembly of coagulation complexes. We have previously shown that increased microvesicle-bearing phospholipid procoagulant activity in the plasma of patients with SCD contributes to accelerated velocity and an elevated peak of TG [39].

These results were observed both with and without thrombomodulin. This protein sensitizes TG to the protein C/protein S anticoagulant pathway and allows a more global exploration of the coagulation balance. Therefore, we observed an extremely low RETP before HSCT which emphasizes the prothrombotic profile resulting from the impairment of the protein C and protein S anticoagulant pathway.

Furthermore, the altered hemostatic balance before HSCT seems to be linked to a hemolysis-endothelial dysfunction phenotype, as suggested by the significant correlations previously reported between markers of hemolysis and parameters of TG in children with SCD [39]. This can be supported by the low Hb and the high LDH level before HSCT.

Our results indicate a clear trend toward the resolution of the hypercoagulable state in SCD as soon as 3 months following successful allogeneic HSCT, as evidenced by the reversal of the kinetic parameters of TG to normal values. This is likely the result of the replacement of sickle red cells by engrafted non-sickle erythrocytes, leading to a decreased level of circulating erythrocyte-derived phosphatidylserine-positive microvesicles. As stated earlier, the level of these microvesicles have been shown to be a critical determinant of the propagation phase of TG in patients with SCD [39]. Interestingly, the improvement of the kinetics of TG persisted over time and was sustainable up to 15 months after engraftment. In parallel, Hb increased, while HbS and LDH level decreased after HSCT and tended towards normal values.

Some TG parameters—lag time, ETP and RETP for instance—did not return to normal values. RETP improved after HSCT suggesting a partial recovery of the impaired protein C/protein S anticoagulant pathway [40]. The reasons for these findings are unclear. Since the study was performed in a pediatric population, we could not get enough sample volume to measure other coagulation parameters, including individual coagulation factors, physiologic anticoagulants, D-dimers and microvesicles. These will be addressed in further studies with the aim to better understand the underlying mechanisms of our observations. Nonetheless, a persistent inflammatory state following HSCT may lead to circulating tissue factor and an elevated level of FVIII, which are potential factors that can contribute to the maintenance of a short lag time, high ETP and reduced RETP in these patients.

Our study has some other limitations. The first one is that it was only centered on SCD children. Indeed, HSCT in SCD is mainly performed during childhood. Our observations might be extendable to SCD adults as they have a similar procoagulant profile. However, the impact of HSCT on the hemostatic system of adults could be different. The second limitation is that the population of the NC was not entirely race matched as it also included Caucasian children. It is well recognized that African people have a higher incidence of thromboembolism as compared to “white” people [41,42]. Roberts et al. reported that African-Caribbean adults have a higher TG profile than Caucasians, with higher peak, ETP and velocity [43]. Despite the fact that lag time, time-to-peak and ETP reduction were not significantly different in Roberts’s study, this highlights the fact that TG studies are influenced by ethnicity. In further studies, we intend to use sibling donors as potential controls. Nevertheless, it is a significant challenge to get samples from normal controls in children for coagulation studies.

Finally, our study is monocentric and concerns a small cohort of patients who all had a successful MSD-HSCT after myeloablative conditioning with full donor chimerism. Our results may not be extrapolated to patients who develop mixed chimerism after transplantation.

## 5. Conclusions

To summarize, a successful HSCT reverses accelerated coagulation reactions in children with SCD. No thromboembolic events were recorded in the patients studied. However, further properly designed studies are essential to evaluate the clinical impact and the underlying mechanisms of the changes induced by HSCT on coagulation in SCD.

## Figures and Tables

**Figure 1 jcm-08-01796-f001:**
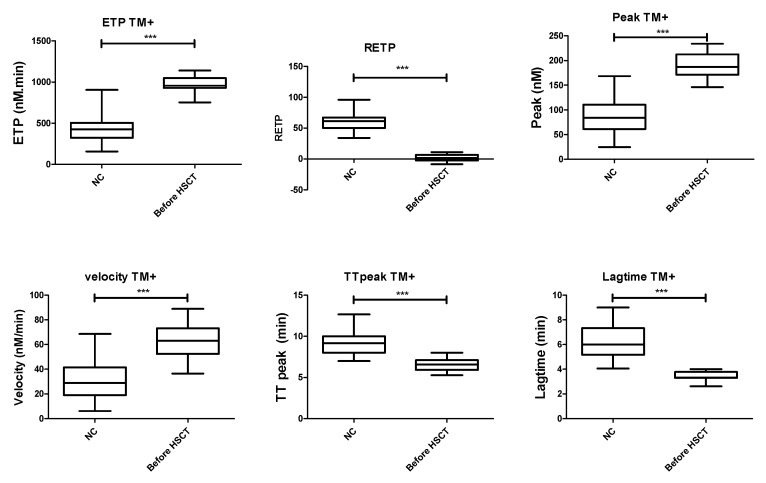
Comparison of the TG parameters with TM (median and range) between normal controls (NC) and sickle cell disease (SCD) patients before HSCT. ETP TM+: endogenous thrombin potential following addition of thrombomodulin, RETP: reduction of ETP with the addition of thrombomodulin, HSCT: hematopoietic stem cell transplantation, TM+: with addition of thrombomodulin, TT: time to peak, NC: normal control, *** *p* < 0.001 as compared to the controls using the Mann–Whitney test.

**Figure 2 jcm-08-01796-f002:**
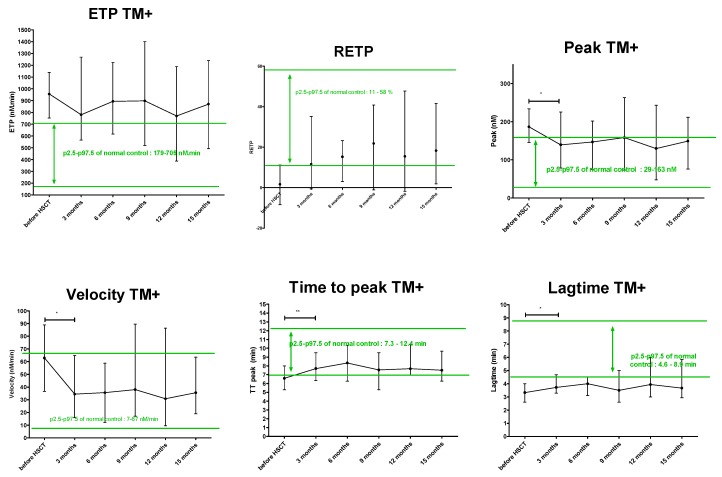
Modification of TG parameters (median and range) before and after successful HSCT, with (TM+) the addition of thrombomodulin. The range p2.5–p97.5 of NC is colored in green. ETP TM+: endogenous thrombin potential following addition of thrombomodulin, RETP: reduction of ETP with the addition of thrombomodulin, HSCT: hematopoietic stem cell transplantation, TM+: with addition of thrombomodulin, NC: normal control, *: *p* < 0.05, ** *p* < 0.01.

**Table 1 jcm-08-01796-t001:** Demographic data of the 17 patients, their donors, and clinical and biological data following hematopoietic stem cell transplantation (HSCT).

**Genotype patient**				16 HbSS, 1 HbSB+			
**Sex**				5 females, 12 males			
**Age at transplant (years)**				9.6 (3.6–16.5)			
**Donor Genotype**				11 HbAS, 5 HbAA, 1 HbAC			
**Donor sex**				7 females, 10 males			
**HSCT complications**	**Acute renal insuffisency**			1, resolved 3 months after transplant, due to drug toxicities)			
	**Hemorrhagic cystitis**			1, with mild hemorrhagic cystitis resolved less than1 month after transplant			
	**VOD**			0			
	**Acute GVHD**			0			
	**Chronic GVHD**			1, with no more immuno-suppressive treatment 12 months after transplant			
	**Before HSCT**	**6 months after HSCT**	**15 months after HSCT**	**Friedmann test**	**Dunn’s post-test**		
					Before versus 6 months after HSCT	Before versus 15 months after HSCT	6 months versus 15 months after HSCT
**Hb (g/dL)**	8.3 (7.2–11.4)	11.7 (10.5–14.5)	12.4 (10.9–13.5)	*p* < 0.0001	**	***	NS
**HbS (%)**	70 (26–88)	32 (0–40)	34 (0–42)	*p* = 0.0001	***	*	NS
**LDH**	466 (256–823)	243 (185–488)	235 (177–317)	*p* < 0.0001	**	***	NS

Data are expressed as median and range. VOD: vaso-occlusive disease, GVHD: graft-versus-host disease, Hb: hemoglobin, HbS: sickle hemoglobin, LDH: lactate deshydrogenase, HbSS: patient with sickle cell disease that are homozygote for HbS, HbSB+: patient with sickle cell disease that are compound heterozygote for HbS and B+ tahalssémia, HbAS: sickle cell trait for HbS, HbAC: sickle cell trait for HbC, HSCT: hematopoietic stem cell transplantation, NS: not significant (*p* > 0.05), *: *p* < 0.05, ** *p* < 0.01, *** *p* < 0.001.

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
