# Peer review of "Is There Any Improvement of the Coagulation Imbalance in Sickle Cell Disease after Hematopoietic Stem Cell Transplantation?"

_jcm, 2019, doi:10.3390/jcm8111796_

Round 1

Reviewer 1 Report

The manuscript “Is there any improvement of the coagulation imbalance in sickle cell disease after Hematopoietic Stem Cell Transplantation?” is a single institution cohort study that describes the association between T cell depletion transplantation and immune cytopenias.  The aim of this study was to determine if hemostatic imbalance that was noted pre transplant improved post HSCT. The findings indicated mixed results, peak, velocity and time to peak of thrombin generation restored to normal by 3 months post HSCT but partial reversal of endogenous thrombin potential post transplant.

This is a well-written manuscript that highlights some of the ways that HSCT reverses some of the pathophysiology of SCD General comments:

This study could be improved by including a better review of the literature on the effects of HSCT on coagulation system. In addition, I believe clarity should be given as to what endogenous thrombin potential and thrombin generation represents and why it was chosen as the best indicators of hemostatic imbalance in patients with SCD. The effect of race on the hemostatic imbalance and how we should interprete the result of this study is very important and should be highlighted ( Robert et al. Blood Coagul Fibrinolysis. 2013 Jan; 24(1):40-9). It was stated clearly that the control were main Caucasians but I think this needs to be mentioned in the interpretations of the results. I think for further studies the sibling donors could be recruited as potential control.

Specific Recommendations:
Title:   I don’t think the  words Hematopoietic Stem Cell Transplantation should start with capital letters

Abstract:  It is stated in the abstract that “previous reports suggest that in-vivo T cell depletion

Results: Can you simplify Table 1? I don’t think we need all the timelines unless you think it is important, just Month 3 and 15 months is adequate.  it is difficult to read as it is now. Also I don’t think you need Table 1 and Figure 1. They are both stating the same thing. Maybe change table 1 to a true table 1 showing the characteristics of the patients that were transplanted and any possible complications (such as gvhd, acute renal insufficiency, VOD) through transplant that may affect the hemostatic balance.  The title should state ETP and TG parameters………

Figure 3 does not technically add anything to your main aim and maybe can be summarized in table 1

Reviewer 2 Report

This is an interesting observational study of the reversal of procaogulant changes in sickle cell anaemia after allogeneic transplantation. The findings appear robust, are well-presented and support the conclusions of the authors. 

I would suggest a more thorough explanation of the coagulation methodologies used, their rationale and how they are used, to aid a non-haematologist audience in understanding their significance.

Round 2

Reviewer 1 Report

Revision is acceptable, good job.